# Less is more: Efficient behavioral context recognition using Dissimilarity-Based Query Strategy

**Atia Akram** [1]*, **Asma Ahmad Farhan**[2], **Amna Basharat**[1]

**1** Department of Computer Science, National University of Computer and Emerging Sciences, Islamabad, Pakistan, **2** Department of Computer Science, National University of Computer and Emerging Sciences, Lahore, Pakistan

* i171401@nu.edu.pk

**Data Availability Statement:** Data is publicly available at http://extrasensory.ucsd.edu/.

**Funding:** The author(s) received no specific funding for this work.

## Abstract

With the advancement of ubiquitous computing, smartphone sensors are generating a vast amount of unlabeled data streams ubiquitously. This sensor data can potentially help to recognize various behavioral contexts in the natural environment. Accurate behavioral context recognition has a wide variety of applications in many domains like disease prevention and independent living. However, despite the availability of enormous amounts of sensor data, label acquisition, due to its dependence on users, is still a challenging task. In this work, we propose a novel context recognition approach i.e., Dissimilarity-Based Query Strategy (*DBQS*). Our approach DBQS leverages Active Learning based selective sampling to find the informative and diverse samples in the sensor data to train the model. Our approach overcomes the stagnation problem by considering only new and distinct samples from the pool that were not previously explored. Further, our model exploits temporal information in the data in order to further maintain diversity in the dataset. The key intuition behind the proposed approach is that the variations during the learning phase will train the model in diverse settings and it will outperform when assigned a context recognition task in the natural setting. Experimentation on a publicly available natural environment dataset demonstrates that our proposed approach improved overall average Balanced Accuracy(BA) by 6% with an overall 13% less training data requirement.

## Introduction

Automated human context recognition, such as person's location and their companions, has a wide range of applications. These include automatic recognition of context for elderly persons [1, 2], fall detection systems [3], disease prevention [4], and even quitting bad habits [5].

With the growing trend of using smart devices [6, 7] and the availability of a range of built-in sensors, these devices have become good agents for recognizing users' behavioral context pervasively [8–10]. However, recognizing natural human behavior is challenging due to its diversity and unpredictability in real-world settings. Human activities play a vital role in identifying behavioral patterns [11]. These patterns, being unique to individuals, may help to

**Competing interests:** The authors have declared that no competing interests exist.

**Abbreviations: AL**, Active Learning. 1, 3, 5, 7, 8, 11, 12; **BA**, Balanced Accuracy. 1, 8, 13, 14, 15, 16, 17, 18; **BP**, Best Performance. 16, 17, 18; **CNN**, Convolutional Neural Network. 4, 5; **CP**, Comparable Performance. 15, 16, 17, 18; **DBQS**, Dissimilarity-Based Query Strategy. 1, 3, 8, 11, 12, 14, 15, 16, 17, 18; **EF**, Early Fusion. 14, 15; **HAR**, Human Activity Recognition. 3, 4, 5, 6, 7, 8, 18; **LSTM**, Long Short Term Memory. 2, 5; **PS**, Phone State. 13, 14, 15, 16, 17; **TV**, Time Variation. 14, 15, 16, 17; **WAcc**, Watch Accelerometer. 13, 14, 15, 16, 17.

identify fine-grained information about users' activities with respect to their context, which can be simultaneous, interleaved, or concurrent, depending on their nature. For instance, a person may be lying in bed while reading a book and having tea, which represents a concurrent activity pattern. Despite the potential applications of context recognition, the main challenge lies in the requirement of a human observer in the annotation process. A successful context recognition system should be unobtrusive, less burdensome, and run smoothly without the need to instruct the person to provide feedback. Therefore, research must be conducted in real-world or natural settings, where these applications will ultimately be deployed [8].

Different sensing modalities play a vital role in accurately predicting human behavioral context by combining data from various sensors embedded in smart devices to overcome the challenges. The purpose of combining sensors is to take advantage of relevant sensor data in different contexts [8, 12–14].

### Limitations of prior research

Machine learning techniques have been widely used to recognize users' activities either using shallow learning or deep learning methods. Shallow learning techniques i.e., K Nearest Neighbour, Naive Bayes, and Random Forest are best-suited algorithms for accurately recognizing different human activities. Moreover, Deep learning methods [15, 16] i.e., Convolutional Networks [17, 18] and Long Short Term Memory(LSTM) [19] due to automatic feature extraction and accuracy are more popular. However, both these methods require a large pool of labeled sensor data that is time-consuming, onerous, and not always feasible or reliable [12]. To gather labeled sensor data in real environments, various annotation techniques have been employed, such as self-reported time-use diaries [20] and ecological momentary assessment [21]. However, these techniques have a history of being disruptive and biased, which can have a direct impact on the activities recorded for context recognition tasks. Other techniques for gathering ground truth, such as recording and later audio or video analysis, are becoming more common, but they may pose ethical and privacy concerns when recording people's activities. Hence, there is a need for a human behavioral context recognition system that employs less labeled data.

Despite being able to recognise human behavioral context with vast amount of sensors data available in smart devices, the previous researches have still limitations that need to be addressed. Most of the activity recognitions methods are applied on small set of activities [22–24]. Moreover, the prior research is performed in the controlled environment where users are bound to perform specific list of activities in a limited area [25, 26]. Similarly, ubiquitous devices usage is the part of users daily living rather than body worn devices in which users feel burdensome. All the issues discussed with the previous researches cannot truely depict the human behaviour in real life where users are free to do any activity by using smart devices. Therefore, research has to be done in natural and realistic settings, satisfying four natural environmental setup: natural way of device usage, freedom of device placement, natural environment, natural behavioral of user.

### Motivation

The availability of an array of built-in sensors in daily used devices has potential to capture human behaviour that is applicable in many fields i.e., Aged care [1], disease prevention [4] etc. For instance, in Australia, it is projected that by 2051, people aged 85 and above will account for 9.1% of the overall population, with the number of individuals aged 100 and above increasing from 2, 503 in 2001 to 38, 000 in 2051 [27]. The growing number of elderly individuals has led to a significant increase in demand for care services. The urge for

independent life in the elderly has stimulated the development of ubiquitous systems that monitor them without being burdensome, remind them to take their medications or eat their meals, and notify authorities or family members if something goes wrong [3, 28]. Moreover, preventive interventions for drug addiction have become increasingly important in light of the pandemic, family issues, peer pressure, depression, and financial difficulties faced by young people. Reducing drug usage is one of the main goals of these interventions, which can be made possible through the behavior control of drug users [29]. Thus, accurate identification systems are needed for successful elderly care and behavior control of addicts.

## Contribution

In this paper, we propose a novel Active Learning(AL) based query strategy that overcomes the gap of label requirement in supervised learning. AL is a semi-supervised machine learning approach that requires a small set of labeled data and queries informative samples from a large pool of unlabeled data. Using a small set of labeled data, a model is trained that helps in the query strategy. The purpose of query strategy is to measure the informativeness of unlabeled samples. AL is an iterative process in which query strategy works by selecting only informative samples using uncertainty measure from the pool in each stage, thus reducing the annotation load. We also investigated the impact of temporal data, such as time of day (morning, evening, etc.), on recognizing user behavior in natural settings. To evaluate our proposed methodology, we used a publicly available dataset of users performing diverse activities in a natural environment.

Key contributions of our research work is as follows:

- We proposed and implemented a novel technique i.e., *DBQS* that aims to reduce the user's annotation effort.

- We exploited the use of time-series analysis to find the diverse and unique samples in the data. Empirically, our technique outperformed when selected sample pool is selected using temporal information.

- Proposed *conditional DBQS* that assigns higher weight to the unique samples. Intuitively, conditional *DBQS* reduces the annotation load by recognizing users' diverse behavior.

- Investigate the mutual influence of temporal information and conditional *DBQS* on sensor data to build an accurate model with less annotation load.

The subsequent sections of the paper includes related work, proposed methodology, experimental evaluation, discussion and conclusion.

## Related work

Smart devices with built-in sensors have the potential to capture useful information for understanding human behavior [8, 9, 12]. However, several factors such as data collection methods [30–33], feature selection [34, 35], sensing modalities [34], and machine learning models [35, 36] can impact the performance of such systems. In this section, we will discuss the state-of-the-art in the domain of Human Activity Recognition(HAR) using smart devices. Firstly, we will discuss the significant work done using supervised machine learning approaches. Next, we will examine the significant semi-supervised approaches used in the domain. Finally, we will also discuss the impact of data collection in both natural and controlled settings on HAR.

## Supervised approaches

Most of the previous research in the domain of HAR [24, 37–39] is based on recognizing basic physical activities like sitting, standing, running, walking, etc. To improve the performance of these recognition systems, context information is then combined with these activities i.e., Sitting at Home/Office, Sitting in a Bus, Sitting in Train, Standing at Home/Office, etc. In this regard, Otebolaku et. al. [35] collected data from 6 individuals in the controlled setting using accelerometer, orientation, and rotation vector sensors of user wearable mobile devices. Different machine learning techniques are applied to analyze the performance and demonstrate an overall performance of 98% using cross-validation. Similarly to get the advantage of contextual data, a unique activity-aware human context recognition(AAHCR) technique is proposed by [36]. Their proposed approach learns human activity routines in various behavioral contexts and predicts user contexts. Daily living physical activities, e.g. sitting, lying, walking, etc., are inferred along with variety of possible contexts, i.e., meeting, in a car, surfing the internet, or lying on bed. They evaluated their proposed approach using a list of machine learning models, of which Random Forest achieved the best recognition rate in recognizing various behavioral contexts, showing their proposed method's effectiveness. Moreover, Niemann et. al. [22] proposed a method to identify activities for industrial processes, and their experiments on a laboratory setup showed that incorporating context information significantly improves activity detection performance. Bhandari et. al. [23] created a lightweight and precise system on smartphones to detect human activity like standing, sitting, laying, walking, walking upstairs, and walking downstairs.

Moreover, a sensor fusion technique is applied by Vaizman et. al. [9] that concatenates data from multiple sensors to improve the overall context recognition accuracy. The data was collected in natural setting from various smartphone and smartwatch sensors. They have used logistic regression, a linear classifier, and the results demonstrated that fusion of multi-modal sensors helps in improving the recognition of user behavioral context in the wild. Zhong et. al. [34] investigated the contribution of mobile phones or smart watches sensors towards machine learning based models for HAR. In real-time environments, it can be challenging to obtain labeled data necessary for supervised learning techniques, particularly in the context of HAR. Consequently, it becomes necessary to explore unsupervised learning techniques using unlabeled data [40].

Deep learning has achieved some success in recent years, by modeling high-level abstractions from complex data. These techniques have demonstrated success in many areas such as computer vision [41], image classification [42], natural language processing [43], and speech recognition [44]. These mostly depend on feature extraction techniques [45] such as statistical methods [46], time-frequency transformation [47], and symbolic representation [48]. Next, we are listing some related articles on HAR using deep learning methods and then AL integrated with deep learning.

The majority of deep learning research aimed at tackling issues associated with HAR data, such as accuracy, class imbalance, hyper-parameter configuration, automated feature extraction, computational and memory costs, employs Convolutional Neural Network(CNN). In a work proposed by Lee et. al. [49], inertial and ambient sensors data are fused to address class imbalance problem using Deep Convolutional Neural Network (DCNN). Evaluation and analysis of their proposed system showed an improvement of 5.3% in accurate recognition. Similarly, Hammerla et. al. [50] analyzed different deep learning models i.e. Deep neural network, convolutional network and recurrent neural network with the perspective of hyper parameters. In more than 4, 000 experiments they investigated the suitability of each model for different tasks in HAR. Further, they also explored the impact of each model's hyper parameters on performance, and proposed guidelines for the application scenario.

Furthermore, Yang et al. [17], automated the feature extraction through CNN. The multi-channel time series data, which primarily uses convolution and pooling techniques to capture the key patterns of the sensor signals at various time scales, is also investigated using CNN. They showed that the proposed CNN method outperforms other state-of-the-art methods and can be used as a competitive feature learning and classification tool for the HAR challenges. Similarly, Khan et. al. [51] created a hybrid model for activity recognition by combining CNN and LSTM. LSTM network is used to learn temporal information, whereas CNN is used to extract spatial features. They conducted a thorough ablation study, developed a new dataset and created a novel hybrid model powered by deep learning to track and recognize human physical activity in an indoor environment. The CNN-LSTM approach yields an accuracy of 90.89%, demonstrating the suggested model's suitability for HAR applications. Another research [25] was conducted in the similar manner by hybridizing CNN with a bidirectional (1D-CNN-BiLSTM) model for wearable sensor-based HAR and reported good performance as compared to existing methods. Recently, [52] designed an accurate and efficient MLP architecture with a stack of dense layers to recognize six daily human activities. Their study used small and large feature sets of gyroscope and accelerometer sensors for HAR and their results outperformed LSTM, CNN, and other machine-learning models.

One of the major challenges of HAR model is to develop a highly accurate model with low cost sensors [19] or less sensor units [53]. A Bidirectional-Gated Recurrent Unit-Inception (Bi-GRU-I) model was created by Tong et al. [53] to increase accuracy while minimizing the number of parameters. Their testing using datasets from the University of California, Irvine (UCI-HAR), Wireless Sensor Data Mining (WISDM), and CATP, which they self-collected, demonstrates improved performance and robustness. Moreover, the sensor configuration optimization was examined, and it was shown that this approach may be used with smaller sensor units. Similarly, in [19], LSTM Networks are employed to enable the model to understand and recognise the various activities that the user is carrying out. The UCI-HAR [54] dataset is used to train the model to recognise and identify six different activities, including sitting, walking, lying down, standing up, going up stairs, and going down stairs. The complexity of updating each weight is lowered to O(1) using LSTM. Training becomes simpler and produces great accuracy as weights are updated more quickly.

All the discussed literature on shallow and deep models demand a wide range of annotated data. Besides the advantage of good performance and automatic feature extraction, the main drawback of deep models is computational cost than the conventional shallow learning approaches. Despite being able to collect sensors data ubiquitously, limited availability of *quality* labeled data remains one of the main challenge [55] in HAR. The issue arises primary due to the fact that data annotation is to be done by the user. Due to this issue, researchers are actively seek solutions that require partial or no labeled data. For this purpose, multiple semi-supervised [37, 56–58] and unsupervised [57, 59, 60] approaches are proposed.

## Semi-supervised approaches

Semi-supervised approaches are the broad category of machine learning techniques between supervised and unsupervised learning i.e., self-training, co-training and active learning etc. Semi-supervised learning is a strategy for training models that uses both labeled and unlabeled input and is employed when just a small amount of labeled data is available. For example, in AL [61], model is trained using small portion of labeled data and gradually quality of unlabeled samples are measured so that to be added in the labeled set. The primary idea underlying semi-supervised learning is the labeling of only a small sample of data that may result in the same or greater accuracy than completely labeled training data. The only challenge is finding

what that sample is in gaining good accuracy. AL is all about labeling data dynamically and gradually during the training phase so that the algorithm can determine which label is the most informative to learn from it. The use of AL in HAR has been studied extensively either in a pool-based setting(offline) or a stream-based(online) setting. With a pool-based approach, the model has access to all unlabeled data and picks the best from it. Stream-based approach, on the other hand, progressively reads unlabeled data and selects the informative one to be queried by the oracle or user. To enhance the performance of an initial classifier, Longstaff et. al. [37] investigated various active and semi-supervised learning approaches to label the activity data stream. For this, they compared AL and three different semi-supervised learning methods i.e., self-learning, democratic co-learning and En-Co-Training. Although democratic co-learning was nearly as effective and did not require user interaction, but AL showed the greatest improvement. In [62], a semi-supervised auto labeling approach is proposed with the key intuition that labeling, being burdensome on user, could lead to a high misclassification rate. Multiple factors like hectic working/living conditions, misleading emotional fluctuations etc. can possibly result in incorrect labeling. For experimentation, they used dataset collected in the natural setting i.e., [30] and [63]. To improve accuracy, lessen overfitting, and solve imbalanced class problem, a special combination of augmentation, scaling, and boosting is applied.

AL is used by System Maestro [26], Liu et al. [64] and Shahmohammadi et. al [24] to reduce annotation load. System Maestro [26] is a data collection and labeling framework that was developed to work in a smart ambient environment with 5 different ambient sensors. AL was employed to achieve accuracy more than 95%, with less than 10% labeled examples. Moreover, Liu et al. [64] presented early investigation of pool-based AL using multi-sensor data of daily physical activities in a controlled setting and demonstrated the effectiveness of AL approach as compared to supervised approaches. Shahmohammadi et. al. [24] described a smartwatch-based AL method in stream-based setting to identify 5 commonly performed daily activities and revealed that smartwatches has significant advantages over smartphones and other devices for activity recognition tasks. NOHAR(NOvelty discrete data stream for Human Activity Recognition) [65] is an attempt to address the issue of computational resources using AL. Their experiments in the controlled setting proved that NOHAR can cut memory usage by an average of 99.97% and 13 times efficient than the state of the art.

AL with the advantage of less input labeled data requirement has been used with deep models to gain the strength of deep learning. The research combined AL with deep learning e.g., DeActive [66, 67], ActiveHARNet [68] proposed a deep and AL-enabled activity recognition model, and their experimentation on real-world data showed optimum accuracy using less labeled instances. Bettini et. al. [69] used AL with federated learning to address the issue of data scarcity, proactively annotation of unlabeled sensor data and construction of personalised models.

The above discussed literature on deep learning and AL integrated deep learning still have high computation cost when compared with shallow models. Moreover, they are being experimented on controlled setting with few activity set and number of participants that may fail while experiencing in the real world setup. To avoid complexity, simple AL technique is found in many articles. Comparative analysis of related articles under controlled and natural setting are shown in Table 1.

Adaimi et. al. [12] investigated AL to reduce annotation load in predicting users' activities in natural and controlled settings. The intention of using AL is to generate a HAR model with high accuracy and less labeled data requirement. Using four separate publicly available HAR datasets collected in different environments, they tested existing AL techniques in pool-based and stream-based settings. They proposed a cluster based diverse query strategy that considers informative samples from the pool. This maintains the data class ratio in each batch for each

**Table 1. Comparative analysis of literature on Human Activity Recognition under controlled and natural setting using different machine learning approaches.**

| Ref./Year | Env (C or N) | DataSet | # of Acts/# of Subjs | Sensors | ML (SP or AL) |
|---|---|---|---|---|---|
| [25], 2022 | C | UCI–HAR [54], Motion Sense [70], Single Accelerometer [71] | 6/30,6/24,8/15 | Acc,Gyro | SP |
| [72], 2022 | N | Extrasensory [30] | 37/60 | Mobile Acc and WAcc | SP |
| [34], 2022 | N | Extrasensory [30] | 5/60 | Acc, Gyro, Loc, WAcc, PS, Aud | SP |
| [22], 2022 | N | oMoCap [73] | 6/2 | Optical markers | SP |
| [23], 2022 | C | WISDM [74] | 6/29 | Acc, Gyro | SP |
| [26], 2021 | C | Self | 3/5 | Ambient Sensors | AL |
| [62], 2021 | N | [63], Extrasensory [30] | 4/19, 60 | Acc, Gyro, Mag and GPS | SP |
| [65], 2021 | C | UCI–HAR [54], The Shoaib database [75], WISDM [74] | 6/30, 8/10, 6/36 | Acc, Gyro, Mag and GPS | SP and AL |
| [12], 2019 | C and N | Extrasensory [30], Opportunity [31] | 22/60 | Acc, WAcc, Gyro, Loc, Aud, PS | AL |
| [57],2013 | C | Opportunity [31] | 15/4 | Acc | AL |
| [37], 2010 | C | Self | 3/32 | GPS and Acc | AL |
| [64], 2010 | C | MIT [32] | 20/20 | Acc | AL |
| [76], 2017 | C | Self | 7/10 | Motion sensors, Comp and Acc | AL |
| [24], 2017 | C | Self | 5/12 | Acc | AL |
| [9], 2016 | N | Extrasensory [30] | 25/60 | Acc, Gyro, Loc, WAcc, Aud, PS | SP |
| [35], 2016 | C | Self | 10/6 | Rotation vector, orientation, Acc, GPS, light, Mic and Temp | SP |

Table Notes: Env for Environment, C for Controlled, N for Natural, Acts for Activities, Subjs for Subjects, ML for Machine Learning, SP for Supervised, AL for Active Learning, Acc for Accelerometer, WAcc for Watch Accelerometer, Gyro for Gyroscope, PS for Phone State, Aud for Audio, Comp for compass, Mag for Magnetometer, Loc for Location, Temp for Temperature, Mic for Microphone

iteration. It also ensures diversity in each batch and takes few most informative samples thus inducing promising results. However, this approach guarantees diversity in each batch but introduces stagnation problem that may arise when similar samples might be a part of a batch, thus not improving recognition performance.

In recent studies, the query strategies like maximum entropy, Query by Committee(QBC), random sampling, and cluster-based diversity [12] have not been shown to perform well for human context recognition w.r.t. balanced accuracy because of users' diverse behavior. Naturally, users' behavior is diverse from each other but their actions are similar in several situations [62]. Activities of different people exhibit a certain level of similarity because of biological features (e.g., gender, height, weight, etc.), the physical environment (e.g., where the subjects move about), or even sensor biases [77]. This behavior of users inspired us to formulate the query strategy that helps in finding the most informative samples that reduce the annotation effort detailed in section DBQS: Dissimilarity-Based Query Strategy.

All these researches on HAR using AL are based on a controlled setting with few activities and a small number of participants. Compared to previous work, in this paper, we present a more thorough investigation of pool-based AL on a larger dataset of user contextual activities from multi-sensor devices i.e., smartphone and smartwatch in the natural setting.

## Proposed methodology

The goal of this research work is to propose an optimal model that uses less data to train, yet can predict human behavior with higher accuracy. To achieve this goal, we proposed the pipeline as shown in Fig 1. In the first step, Data Collection phase acquired the data ubiquitously

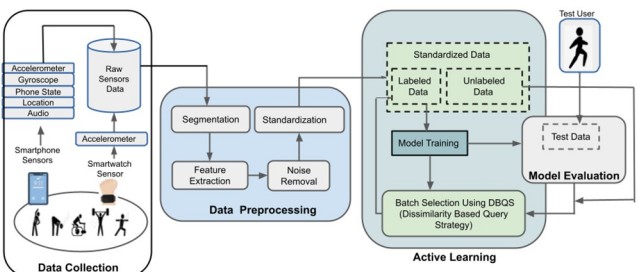

**Fig 1. High level architecture for behavioral context recognition using Active Learning.**

from the sensors embedded in the user-owned smartphones and smartwatches. The smartphone sensors involved are an Acc and a gyroscope etc. Similarly, watch includes sensors like Acc, compass, etc. The raw data acquired from the embedded sensors are, pre-processed in the next phase. The pre-processing steps involved are segmentation, feature extraction, noise removal, and standardization to check for missing values, noisy data, and other inconsistencies and to convert the raw data into meaningful form. Thirdly, we used AL to iteratively learn the users' behavioral context in the natural setting [24]. AL is a type of machine learning algorithm that begins with a small set of labeled data and gradually incorporates meaningful samples from unlabeled data while retraining the model until convergence. To evaluate the performance of the proposed AL method i.e., *DBQS*, BA is employed to quantify the output of the model in each successive iteration. In the following subsections, we will cover the details of each of the steps outlined.

## Data collection

Human behavior is influenced by multiple factors like environmental, physiological, and psychological factors. One of the recent trends in understanding human behavior is the use of smart devices like smartphones and smartwatches. In this work, we acquired the data collected from the sensors of the smart devices like phone accelerometer (pAcc), phone gyroscope (pGyro), and watch accelerometer (wAcc). Eq 1 summarizes the data collected from the smart devices.

$$I = \{X, Y\} \tag{1}$$

where $I$ is the input instance, $X$ represents the raw input sensor data, and $Y$ represents the ground truth information. Specifically,

$$X = \{pAcc, pGyro, pAud, pLoc, pPS, wAcc, pTS\} \tag{2}$$

and

$$Y = \{actLabel, actEnv\} \tag{3}$$

where in Eq 2, *pAcc, pGyro, pAud, pLoc, pPS, wAcc* represent raw sensor readings from the smartphone's Acc, gyroscope, audio, GPS, state, and smart watch's accelerometer respectively, collected at a certain time stamp *pTS*.

Whereas, in Eq 3, *actLabel* refers to physical activity like running, walking, standing, etc. and *actEnv* refers to the environmental information like: outdoors, at the workplace, at home, in gym, etc.

Our proposed system takes $N \times d$ dimensional data, where $N$ represent number of samples and $d$ is sum of all sensors axes representing the columns of data $X$ as shown in Eq 4:

$$X \epsilon R^{N \; x \; d} \tag{4}$$

while $Y$ represents the corresponding ground truth information, i.e.:

$$Y \epsilon \{0, 1\}^{N \times L} \tag{5}$$

In Eq 5, $N$ is the number of samples and $L$ is the label in the form of binary representation where 1 and 0 means if a particular activity being performed or not performed respectively.

In this work, we intend to take into account several actions occurring at once. For instance, "Walking, Outdoor, Shopping, Phone on desk" or "Sitting, in car, Phone in hand, Talking, With friends". Thus, input to our proposed system will be a single instance of all sensor readings along with multiple activities or labels.

However, there are some challenges associated with the acquisition of data in a real-time environment from multiple sensors e.g.,

- Each sensor's raw data acquired from smart devices is in a different format e.g. Acc, Gyro, WAcc [x,y,z], Loc [longitude, latitude], and Audio is in the form of msec frames and PS is of discrete type. All these types of raw data are not useful.

- The raw data generated by different sensors may contain various types of noise. Therefore, signals are pre-processed before classification to eliminate the unwanted noise from raw data. [78].

- All the sensors have different sampling rates i.e., the rate at which the samples are taken, may lead us to inaccurate human behavior recognition due to biasness in sensing frequency.

- All the sensors are not available at the same time, e.g., the user might remove the watch while sleeping, turned off the location service, or during a phone call audio was not available (to preserve privacy) [8].

These challenges lead us to preprocess the data so that accurate human behavior recognition is possible. Next subsection will outline the methods used.

## Data pre-processing

As data were collected from real world stream of users performing multiple activities in the wild, there is a clear chance of noise that requires preprocessing [79], as a good processing mechanism guarantees a good recognition. All data preprocessing steps are described in the next subsections.

**Data segmentation.** As raw data is coming in the form of a continuous stream with a variable sampling rate, we used a data segmentation technique that allows us to segment the data into fixed window sizes which can be of two types: (1) overlapping windows, in which time windows intersect and (2) non-overlapping windows, in which time windows do not intersect. Since the input to our system is time-series data, to segment we used a non-overlapping sliding window as it takes less time than an overlapping sliding window [80], and segment the data into non-overlapped chunks of $n$ time duration.

**Feature extraction.** The smart devices generate signals of raw sensors with different axes e.g. Acc, Gyro and WAcc are 3-axial, audio raw data is for every few msec frame, location is in the form of latitude, longitude, and PS features are binary indicators, specifying details like app-state, WiFi connectivity, and time-of-day.

After segmentation, each window of raw sensor data is converted into a useful feature vector to make the data meaningful. A feature vector is the main component in any recognition

system and for the proposed method, time-domain features e.g. mean, median, entropy, etc. and temporal features e.g. morning, evening, etc. were used for their simplicity.

After applying segmentation and feature extraction on the raw data, the Eq 2 is transformed to Eq 6, where $Acc_f$ represent the feature vector $f$ of Acc sensor.

$$X = \{Acc_f, Gyro_f, Aud_f, Loc_f, PS_f, wAcc_f\} \tag{6}$$

**Noise removal.** The inertial sensors data acquired from the smartphone and smartwatch are prone to several types of noise and interference for instance equipment noise or noise produced by the user's unconscious body movements that must be treated to avoid biasness or misleading results before any further processing. To tackle such types of missing values, we ignored the instances for which either sensors data or label(s) is missing as this is the simplest or low-cost approach [36].

**Data standardization.** Data Standardization [81] which aims to uniformize the range of features in the input data set, is a crucial approach that is frequently used as a pre-processing step prior to the application of any machine learning model. We applied *Z-Score* a most common technique for standardising data, which is accomplished by deducting the mean and dividing by the standard deviation from each value of each feature.

$$Z-Score = \frac{(X - \mu)}{\sigma} \tag{7}$$

Eq 7 is the mathematical formulation of the z-score, where $X$ is the input data matrix of the feature set, $\mu$ is the mean and $\sigma$ is the standard deviation of the feature set.

Once the data standardisation is complete, all features will have the same scale, mean of zero, and standard deviation of one. Now, the preprocessed data is finally in the form that can be given as input to the AL framework.

## Active Learning framework

One of the major issues in real-world studies using supervised methods is the demand of enormous amount of labeled training data to reliably learn human behaviour. It has become clear that providing thorough and reliable ground truth labeling for the enormous amount of data takes time and is not always practical or reliable.

Thus in order to get the advantage of only available labeled data we are focusing on using the AL approach. AL's central premise is that, with less training data, a learning algorithm may be able to perform as well as or even better than standard supervised approaches provided it can select the most informative data to learn. Thus, AL model only queries for the data that can iteratively increase knowledge and performance. There are two frameworks proposed in research for AL: (1) pool-based and (2) stream-based. We have only employed pool based approach in this research and every stage of the framework is detailed in the next subsection.

**Pool-based Active Learning.** This type of approach is based on two types of data, (1) Seed: which is a tiny subset of labeled data, and (2) Pool: which is a vast pool of unlabeled data. This AL approach, as shown in Fig 2 starts with a seed of the data for training. The size of the seed is a user-defined hyperparameter. The Machine Learning model is trained on the seed batch, and the AL pool-based selective sampling method is used after the model has been fitted. In a pool-based selective sampling strategy, the pool of unlabeled data is available to the model on which it applies the query strategy. The query strategy chooses a new sampled batch consisting of the best informative samples and add these to the already labeled data. Then model is updated, and the process repeats with the updated data. The sampled batch size is

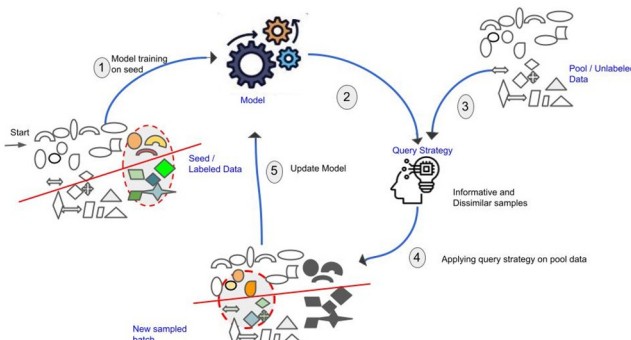

**Fig 2.** *DBQS* **(Dissimilarity-Based Query Strategy): Pool based Active Learning framework.**

also user-defined hyperparameter. The selection of the best samples is decided according to a proposed *DBQS* query strategy as detailed in the following subsection.

**DBQS: Dissimilarity-Based Query Strategy.** Algorithm 1 presents the proposed *DBQS*, an AL approach that overcomes the stagnation problem by introducing a dissimilarity component that prevents repetitive sampling within and outside the batch, thus avoiding redundant labeling problem. *DBQS* consists of two key components: uncertainty and dissimilarity measures for unlabeled samples. To maintain diversity in each successive batch, a new batch is created based on clustering criteria [12].

*DBQS* utilizes margin-based uncertainty sampling [82] to measure informativeness and clustering [12] to preserve the data distribution of the whole training set in each successive batch of each iteration for diversity. Specifically, unlabeled samples are ranked in ascending order by a margin, and each sample from the ranked list is added to the batch while maintaining the same distribution over classes or clusters as the whole training data. This clustering constraint ensures class-level diversity in each successive batch but avoids instance-level diversity in the current and previous batches, which could lead to accuracy stagnation or degradation. Therefore, the dissimilarity component is introduced in the *DBQS* for the selection of unlabeled samples.

The dissimilarity component of *DBQS* rewards samples with high scores when they are dissimilar to previously labeled samples, thereby prioritizing instances that explore unknown parts of the instance space. *DBQS* selects only informative (uncertain) and dissimilar unlabeled samples. In the starting iterations, when labeled data is scarce, the $\alpha$ parameter gives more priority to diversity, while in later iterations, as more labeled data is added, the parameter shifts the priority towards uncertainty.

According to the diversity in sensors data, we extended *DBQS* as conditional *DBQS* by adding a condition in the Euclidean distance statement that captures only the distance of instances with a distance greater than the mean of all distances, thereby giving more weight to dissimilar instances. As a result, conditional *DBQS* only queries instances of dissimilar nature.

**Algorithm 1** Pool-based Active Learning Algorithm

```
Require: Pool data (X), model (M) trained on seed, Size of batch (S),
and number of activities (N_clusters)
Ensure: Indices of selected samples in Batch S
  1: predict_prob ⇐ Class probabilities of X predicted by model M
  2: margin ⇐ Difference between two most probable classes
  3: Distance ⇐ EuclideanDistance (Pool, Seed)
  4: mean = mean(Distance[Distance > 0])
  5: Distance ⇐ Distance > mean
```

```
6:  Similarity ⇐ 1/(1 + Distance)
7:  α ⇐ len(Pool)/(len(Pool) + len(Seed))
8:  Score ⇐ α * (1 − Similarity) + (1 − α) * margin
9:  rank_ind ⇐ argmin(Score)
10: new_batch ⇐ rank_ind with N_clusters distribution
11: return (new_batch)
```

**Classification model.** To train the model on the initial batch, Logistic Regression (LR) [12] algorithm is used. LR is a kind of parametric classification model that have a certain fixed number of parameters that depend on the number of input features, and they output categorical prediction.

$$x = \theta \cdot weight + b \tag{8}$$

In Eq 8, $\theta$ and $b$ are parameters that are obtained using an iterative optimisation algorithm like Gradient Descent or Maximum likelihood. *Weight* are the input observations or features of input data. After calculating the value of $x$ using Eq 8, logistic or sigmoid function is applied.

$$F(x) = \frac{1}{1 + e^{-x}} \tag{9}$$

Eq 9 is a sigmoid function with the property of mapping any real value between 0 and 1. If the value of the input data ($x$) approaches positive infinity, the projected value of $y$ becomes 1; otherwise, $y$ becomes 0. And if the result of the sigmoid function is more than 0.5, we categorize it as class 1 or positive class, otherwise labeled as class 0.

## Model evaluation

To quantify the performance of AL model, we used BA [83] as this suits the dataset because of the imbalanced nature of user activities. BA is the arithmetic mean of the sensitivity and specificity:

$$BA = \frac{Sensitivity + Specificity}{2} \tag{10}$$

$$Sensitivity = \frac{TP}{TP + FN} \tag{11}$$

$$Specificity = \frac{TN}{TN + FP} \tag{12}$$

whereas in Eq 10, BA is balanced accuracy, in Eq 11, TP is a true positive, FN is a false negative, in Eq 12, TN is a true negative and FP is the false-positive case.

## Experimental evaluation

This section presents our experimental evaluation organized as follows: in section Experimental setup, we describe the dataset, selected labels, feature set, classifier, and evaluation approach. Results of proposed method on all sensors with respect to BA and percentage of training data are provided in the result section.

### Experimental setup

We have applied our proposed technique on ExtraSensory DataSet [30] collected by Yonatan and Ellis that contains over 300k examples(minutes) of 60 users performing Daily Living

Activities (DLAs) in the natural setting. The collection includes data from five smartphone sensors (34 iPhone and 26 Android): Accelerometer (Acc), Gyroscope (Gyro), Location (Loc), Audio (Aud), and Phone State sensors (Phone State), as well as accelerometer readings from a wristwatch (WAcc). Raw data collected from smart devices is then segmented into 20 seconds non-overlapping windows that is further processed to create feature vector of each window against each sensor. Further for evaluation purpose, we repeated the experiment with 40 seconds window. Few of the features from ExtraSensory Dataset [30] are listed in the Table 2. Along with these feature sets, we also extracted Temporal features from the time stamp. For this, we generated two combinations: (1) Time Variation 3: A day is divided into three parts i.e., morning, evening, and night with hour wise division as 4 : 00 to 12 : 00, 12 : 00 to 18 : 00, and 18 : 00 to 4 : 00 respectively (2) Time Variation 6: A day is divided into six parts like early morning, morning, noon, evening and night and late night with hour wise division as 4 : 00 to 6 : 00, 6 : 00 to 12 : 00, 12 : 00 to 15 : 00, 15 : 00 to 18 : 00, 18 : 00 to 23 : 00 and 23 : 00 to 4 : 00 respectively. This division is in line of keeping in view of users' diverse behavior during different parts of the day. The dataset contains numerous binary context labels for each occurrence of the selected DLAs, which offer information on the human behavioural context (such as the user's activity, circumstance, location, social context, and phone position) when performing a physical activity in a realistic setting. As a result, we chose this publicly available Extrasensory dataset [30] for developing and verifying the suggested strategy since it fits into the proposed method's process.

The total context labels available in the dataset are 51 and for implementation, we selected 22 frequent occurring labels that include "sitting, lying-down, walking, running, bicycling, sleeping, lab-work, in-class, in-a-meeting, loc-main-workplace, indoors, outside, in-a-car, drive-i-m-the-driver, drive-i-m-a-passenger, at home, restaurant, phone in pocket, exercise, cooking, strolling, bathing-shower". To extend the experiment performed by a cluster-based sampling scheme [12], we applied DBQS. In DBQS, for uncertainty, we used a margin-based uncertainty measure, and dissimilarity is calculated using the inverse of similarity measurement of each new batch with the previously selected data.

More details of the dataset and experimental setup are described below.

The experimental setup comprises two parts: (1) the Single-Sensor classifier and (2) the Early Fusion(EF) approach. In Single-Sensor classifier for each of the 5 sensors—accelerometer (Acc), gyroscope (Gyro), watch accelerometer (WAcc), location (Loc), and audio (Aud),

**Table 2. Feature set of ExtraSensory data.**

| Sensor or Data Category | Feature Names |
|---|---|
| Accelerometer | Mean, Standard deviation, Moment3 and Moment4, Percentile50, Percentile75, Percentile25 etc. |
| Gyroscope | Mean, Standard deviation, Moment3 and Moment4, Percentile50, Percentile75, Percentile25 etc. |
| Location | Log-latitude range, Log-longitude range, Min altitude,Max altitude, Min speed, Max speed, Best horizontal accuracy etc. |
| Phone State | App State, Battery State, Wifi Status etc. |
| Audio | Mean, Standard deviation of mfcc0 to mfcc12 etc. |
| Watch Accelerometer | Mean, Standard deviation, Moment3 and Moment4,Percentile50 etc. |
| Time Variation (TV-3) | Morning Evening and Night |
| Time Variation (TV-6) | Early morning, Morning, noon, Evening, Night and Late night |

phone state (PS) for a given context label, classification is done based on each sensor independently. In the EF approach features from multiple sensors are concatenated before classification. In our experimentation, we used all the features provided in the ExtraSensory Dataset [30] for the mentioned sensors. Both variations of temporal data are concatenated with Single Sensor approach and EF approach separately. For evaluation purpose, we implemented a cluster-based sampling scheme [12] as a base-line method. We used logistic regression with balanced class weights. Further for evaluation we used BA by partitioning the data into the ratio of 80 : 20 thus, 60 users data into train (48 users) and test (12 users) sets. For validation testing, we used In the following experiments, the initial batch size and the pool-based queried batch size were empirically set to 2%.

## Results

The proposed *DBQS* approach discussed in the DBQS: Dissimilarity-Based Query Strategy section relies on uncertainty and dissimilarity measure, therefore, in this section, we have shown results produced from the proposed method with respect to BA and annotation effort in Fig 3. Fig 3 shows the learning curve of the average BA measure over 22 context labels for Acc, Gyro, WAcc, Loc, Aud and PS and EF. Table 3 shows the outcome of proposed method describing percentage of training data need for acheiving a stable BA. Further, for comparative performance of *DBQS* and TV we have shown three-bar plots in Fig 5, for the stabilization wait time (*T*), where *T* = 4, 5, 6 to get Comparable Performance (CP). We have only considered conditional dissimilarity in the case of single sensor classifier and dissimilarity in the case of EF, due to better results as compared to opposite approach.

### Impact of proposed query strategy on balanced accuracy and training data requirement

In the case of Acc, BA gains 59% to 68% with upto 24% training set using 2% batch size and then stabilizes after that. For Gyro, BA of 56% to 66% is gained with 40% training set. Proposed query strategy has no impact on WAcc data due to too much variation in users' behavior in hand motion. Aud sensor data gains BA of 55% to 67%, with upto 24% training set for proposed query strategy. With Loc data, only 1% of increment is observed at the start of few iterations. PS sensor data stabilizes at 73% BA with upto 60% training set using proposed query strategy. In the case of EF, with the use of almost 10% training data, BA stabilizes to 75%.

### Impact of proposed TV on balanced accuracy and training data requirement

In the case of the Acc sensor, for TV-3 and TV-6, BA stabilizes to 68% with only 16% training data and 71% with only 26% training set respectively. For Gyro, for TV-3 and TV-6, 68% and 71% BA is observed with 54% and 60% training set respectively. WAcc with TV-3 and TV-6, BA increases from 57% to 68% with 40% training set and 62% to 72% with 62% training data. For Aud TV-3 and TV-6, BA stabilizes to 69% and 72% with 60% training set. PS sensor showing the same curve of baseline method. Loc sensor shows a prominent increase of 3% and 7% in BA with only 20% training set with TV-3 and TV-6 respectively. EF approach shows the same curve but at some points, BA goes down than baseline method.

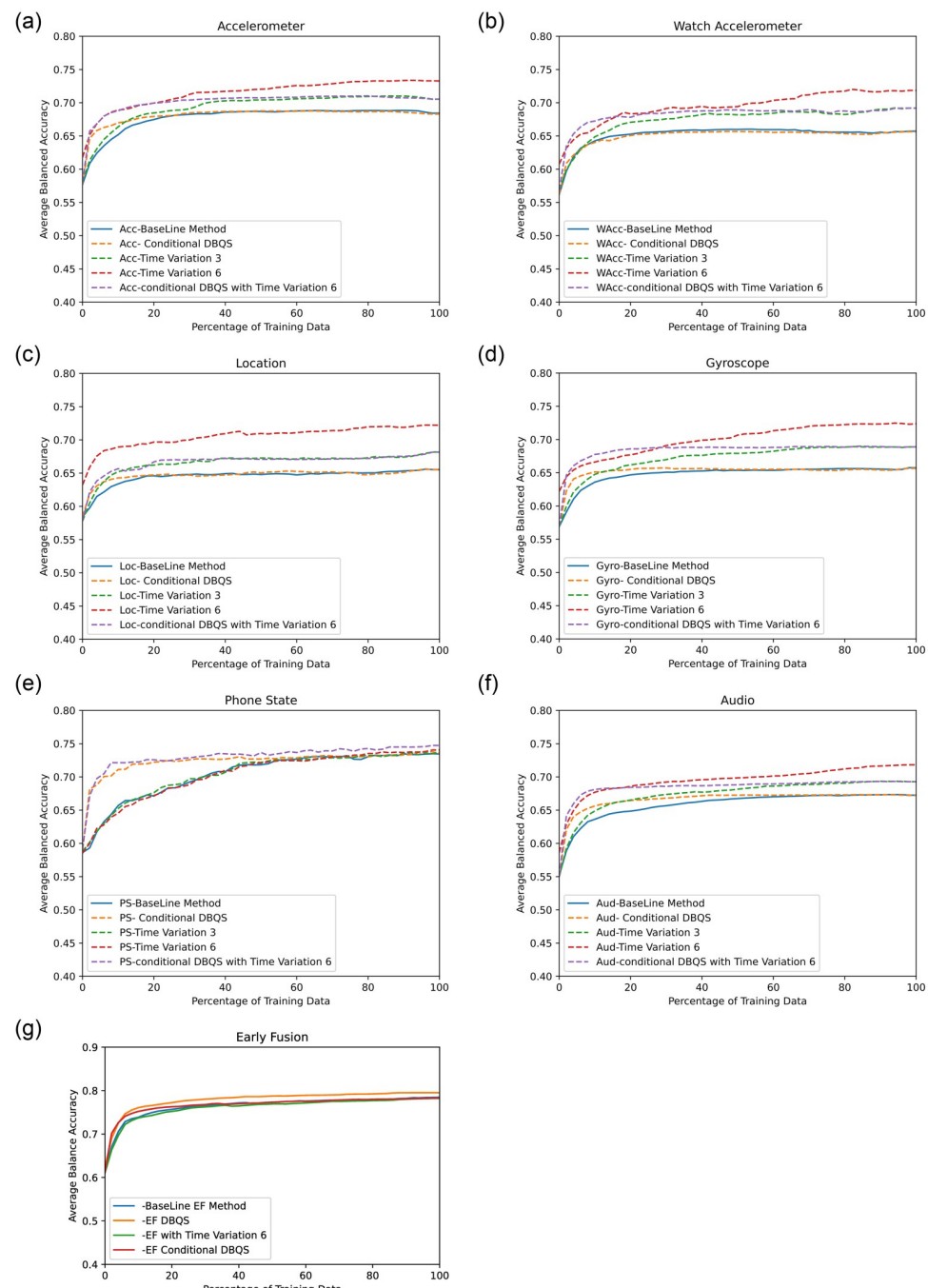

**Fig 3. Results of single sensors and early fusion approach for baseline and theproposed methods (Dissimilarity-Based Query Strategy,Time Variation-3 and Time Variation-6).**

## Impact of proposed query strategy and TV on balanced accuracy and training data requirement

Given that TV-6 yields better results as compared to TV-3, we have only considered TV-6 when combined with the proposed query strategy. When both schemes are applied, the Acc sensor stabilizes after 18% training data with 70% BA. Aud and PS sensor stabilizes after 28%

**Table 3. Results of Dissimilarity-Based Query Strategy (DBQS), Time Variation-6 (TV-6) and Dissimilarity-Based Query Strategy with Time Variation-6.**

| | DBQS | | TV-6 (Time Variation) | | DBQS with TV-6 | |
|---|---|---|---|---|---|---|
| Sensor | Training Data | Balanced Accuracy | Training Data | Balanced Accuracy | Training Data | Balanced Accuracy |
| Acc | 24% | 68% | 26% | 71% | 18% | 70% |
| Gyro | 40% | 66% | 60% | 71% | 22% | 69% |
| PS | 60% | 73% | 42% | 70% | 28% | 73% |
| Aud | 67% | 24% | 60% | 72% | 28% | 69% |
| Loc | 10% | 64% | 20% | 69% | 22% | 67% |
| WAcc | 20% | 64% | 20% | 69% | 14% | 68% |
| EF | 10% | 75% | 20% | 74% | 24% | 68% |

training data with 69% and 73% BA respectively. Loc and Gyro stabilize after 22% training data with 67% and 69% BA respectively. WAcc stabilizes after 14% training data with 68% BA.

## Impact of increased window size on proposed methods results

We have repeated the experimentation for a greater window length of 40 seconds(40*s*) to analyze the final performance. In Fig 4, the method TV-6 and DBQS with TV-6 showing the same recognition pattern against all the sensors. Here we can clearly see the overlapping pattern. However in the case of PS sensor, DBQS is performing better.

## Comparative performance of *DBQS* and TV

The bar chart plotted in Fig 5 shows the Best Performance(BP) and Comparable Performance (CP) of single sensor classifiers when setting $T = 4, 5, 6$ as the stabilization wait time (consecutive number of times when BA stabilizes at a point). As $T$ increases, *CP* gets closer to *BP* because more training data is required to achieve the BP. There is a significant difference in the percentage of training data between *CP* and *BP* as shown in the horizontal axis of Fig 5. This significant difference is the main achievement of our proposed methodology that shows how much annotation load is reduced when less amount of more informative and dissimilar samples are selected.

From the bar plots, we concluded that TV-6 has achieved highest BA. To further visualize the categorization results of the proposed methods, we repeated the experiment with 80% and 20% data distribution. For this, we made different setting by imputing all missing labels to 0 for extracting training and testing data from the whole dataset. The purpose behind this setting is to get same number of instances for all labels to compute confusion matrix. We plotted the normalized multi-label confusion matrices(MLCM) [84] as shown in Figs 6–8, of our TV-6 method against Acc, WAcc, Aud, Gyro, Loc, PS sensor only at the stabilization time ($T = 5$). When setting $T = 5$, all the labels in each sensor converged at different percentage of training data, i.e., in the case of Acc, label *sitting* converged at 30% BA with 20% training data. Due to the highly imbalanced nature of users' activities data diagonal entries not showing good accuracies. Further, a few activity labels i.e., strolling and bathing shower, have not been recognized due to the availability of a single class in this partitioning.

We present a detailed comparative analysis on the annotation load of all sensors when stabilization wait time is set to 6. As compared to base line's BP, *DBQS* along with TV-6 reduces the annotation load for accelerometer, audio, location and gyroscope by 6%, 18%, 8% and 46% respectively. No annotation reduction is found in the case of WAcc. When compared to base line's CP, *DBQS* along with TV-6 reduces the annotation load for Acc, Aud, PS and Loc by 6%,

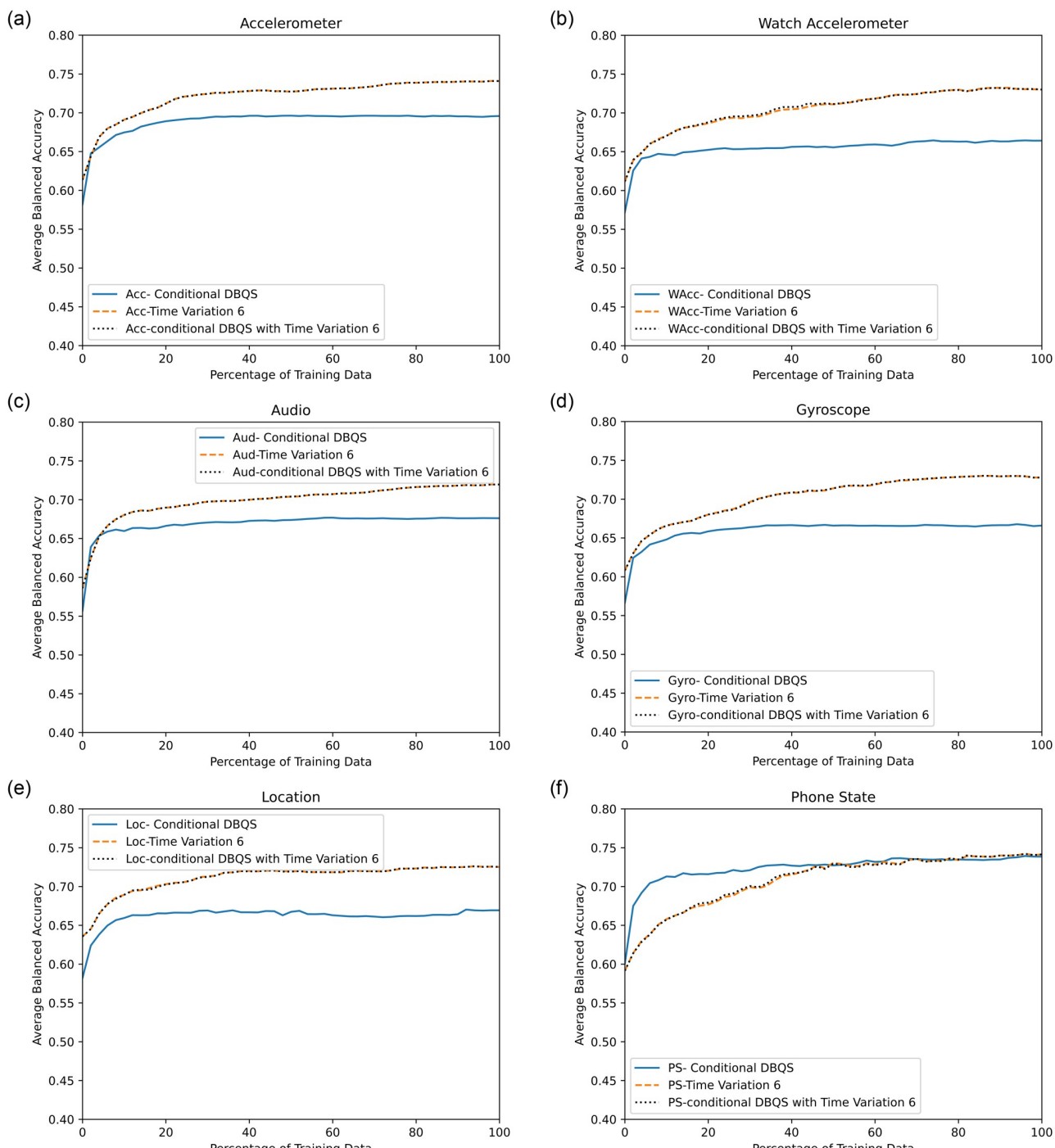

**Fig 4. Results of single sensors approach for the proposed methods (Dissimilarity-Based Query Strategy,Time Variation-6 and combined) with window length of 40s.**

8%, 18% and 4% respectively. No annotation reduction is found in the case of watch accelerometer and gyroscope. For $T = 4$ and 5, almost similar annotation reduction can be observed. Among all sensors, except the PS, TV-6 has high BA that shows more details of temporal data significantly impacts accurate recognition of users' behavior in the natural setting. For

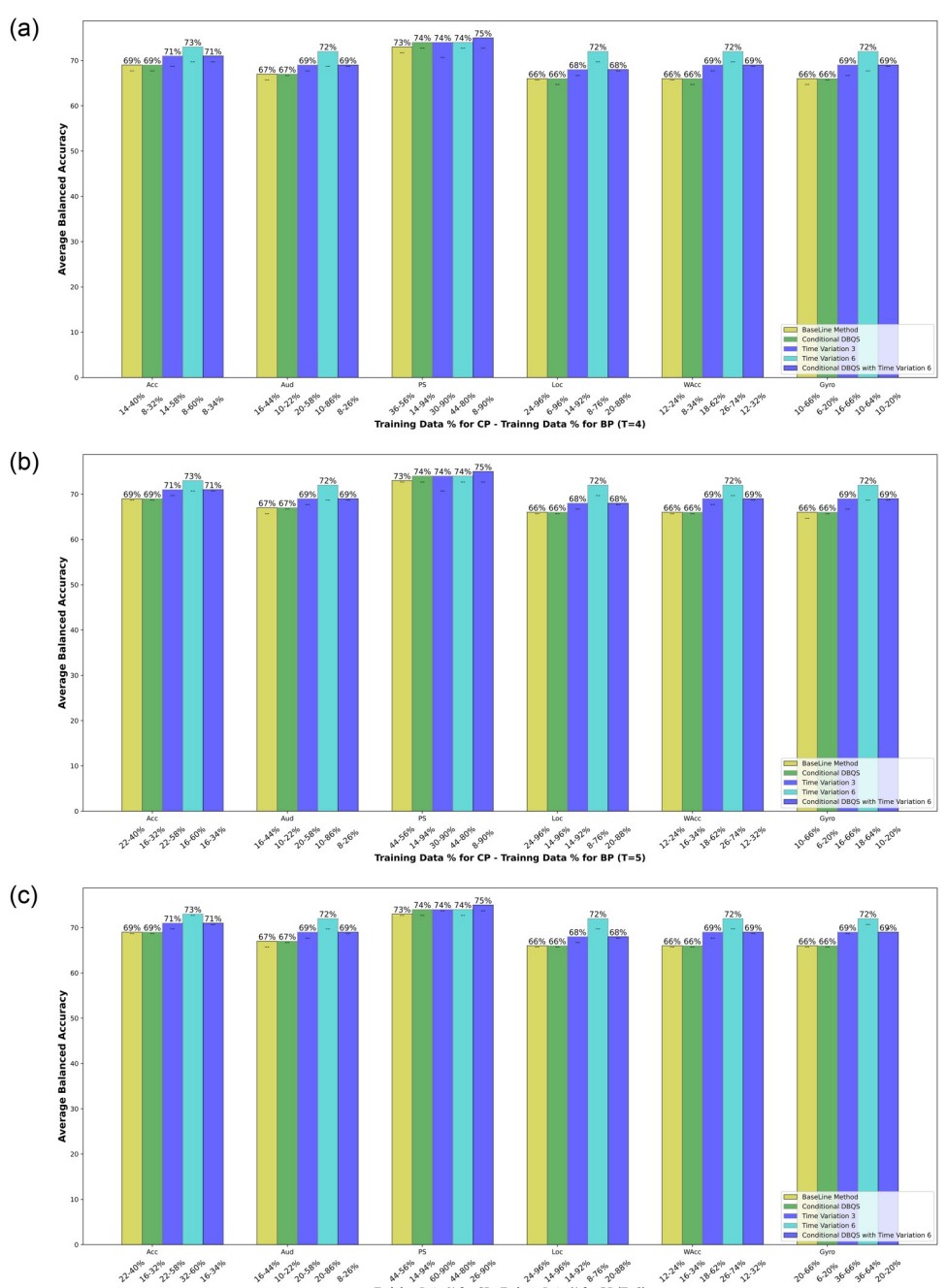

**Fig 5. Comparative performance of Dissimilarity-Based Query Strategy, Time Variation-3, Time Variation-6, Dissimilarity-Based Query Strategy with Time Variation-6 and base line when setting stabilization wait Time $T$ = 4, 5, 6.** The x-axis labels include the percentage of the training data used to achieve the corresponding Comparable Performance when Balanced Accuracy stabilizes and the Best Performance.

example, accelerometer recognises all activities with 73% BA using TV-6 that is 4% better than cluster-based sampling scheme [12].

(a)
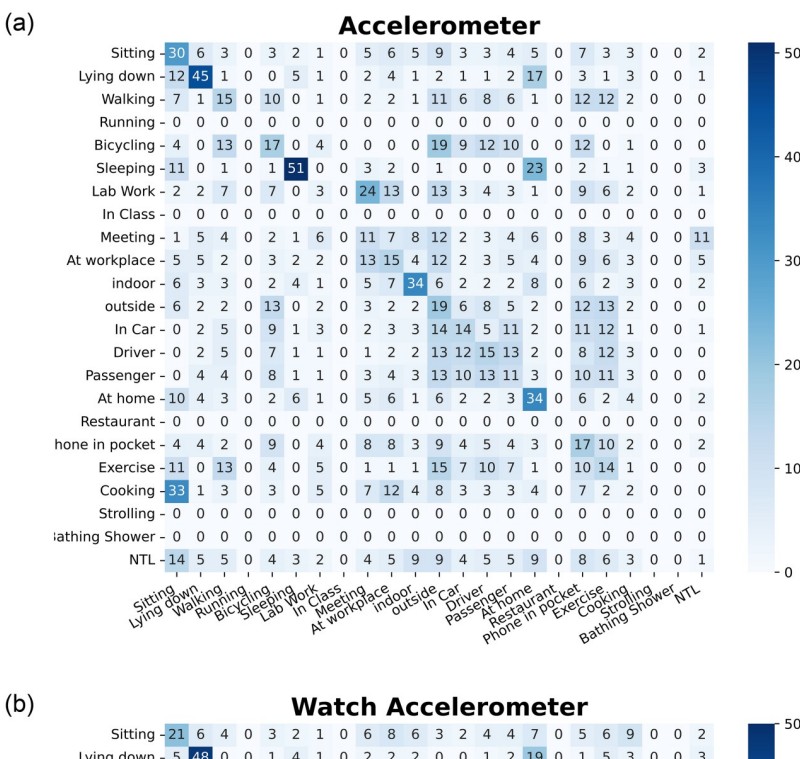

(b)
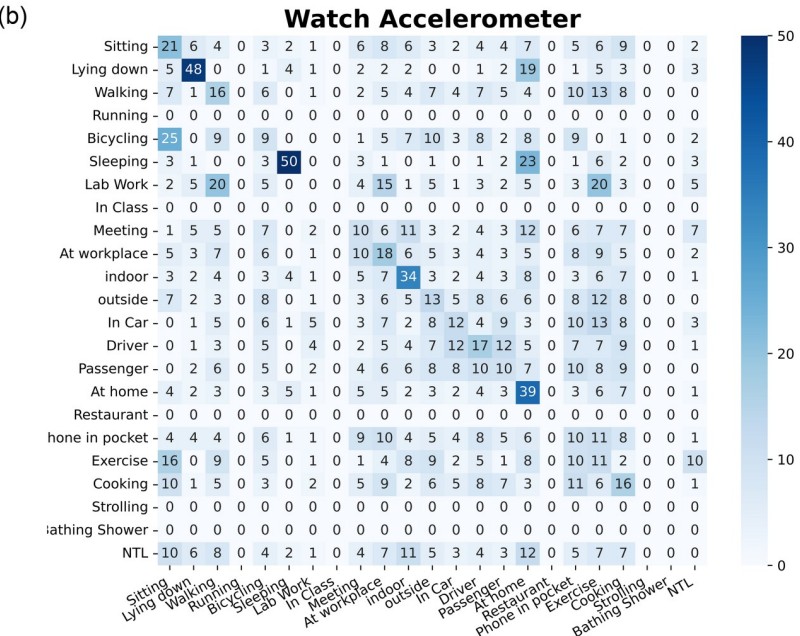

**Fig 6. Confusion matrix for accelerometer and watch accelerometer sensors at thestabilization time, T = 5.**

## Discussion

The proposed methods revealed significant performance (BA and annotation effort) of users' behavior in a natural setting. Almost all sensors showed improved BA as compared to baseline. For example, accelerometer and audio sensors reported 73% and 72% BA, which is 4% and 5% more than the baseline, respectively. Another study [36] of behavioral context recognition in the natural setting, indicated BA of 90% with 100% training set. As compared to their

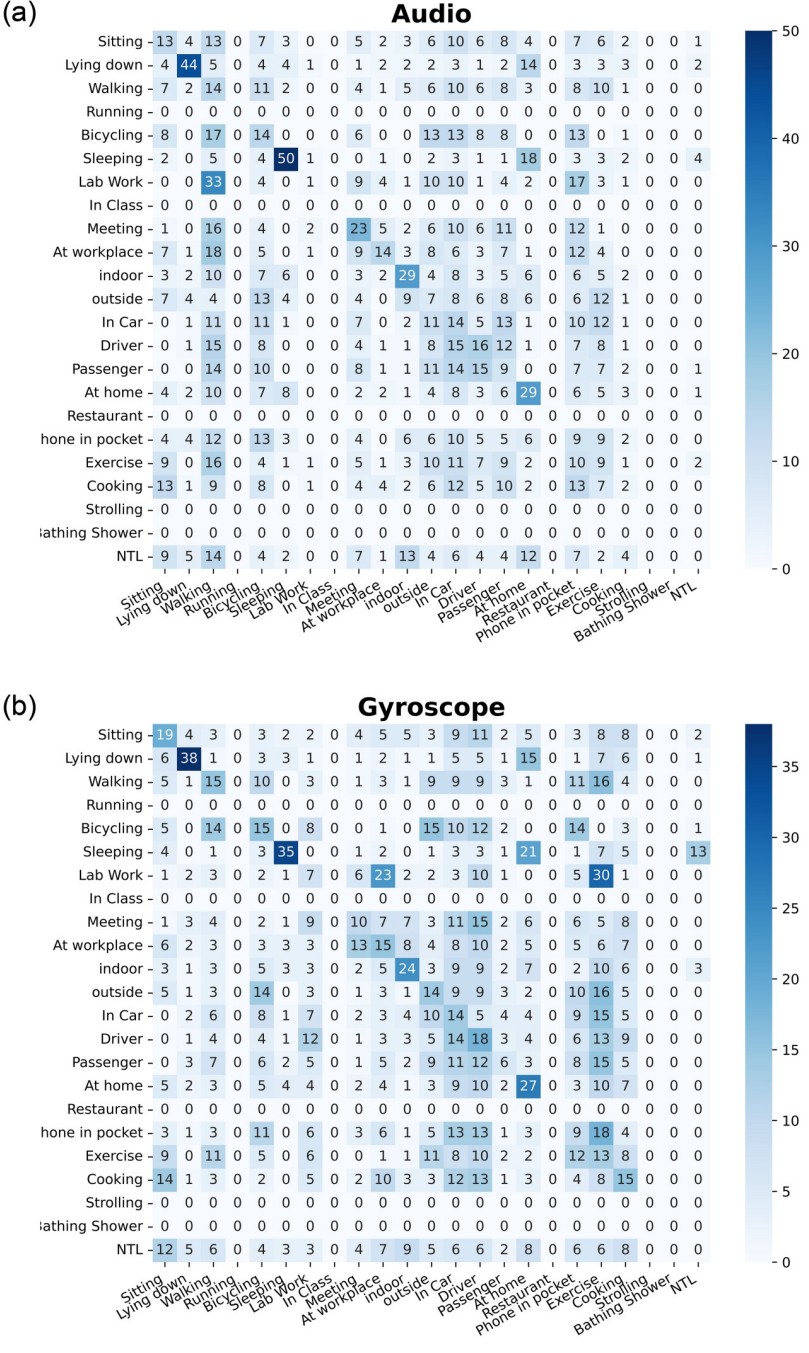

**Fig 7. Confusion matrix for audio and gyroscope sensors at the stabilization time, T = 5.**

approach, we recognized behavioral contexts with 34% training set that is the main advantage of our approach. The reason behind their improved BA is the use of 100% training set that is the main drawback of their approach. Similarly, in our approach location, watch accelerometer, and gyroscope gained 6% in BA with temporal information (TV-6), and we credit this improvement to the temporal information we included with the sensors data. Therefore an improvement in the BA demonstrated the effectiveness of temporal information in

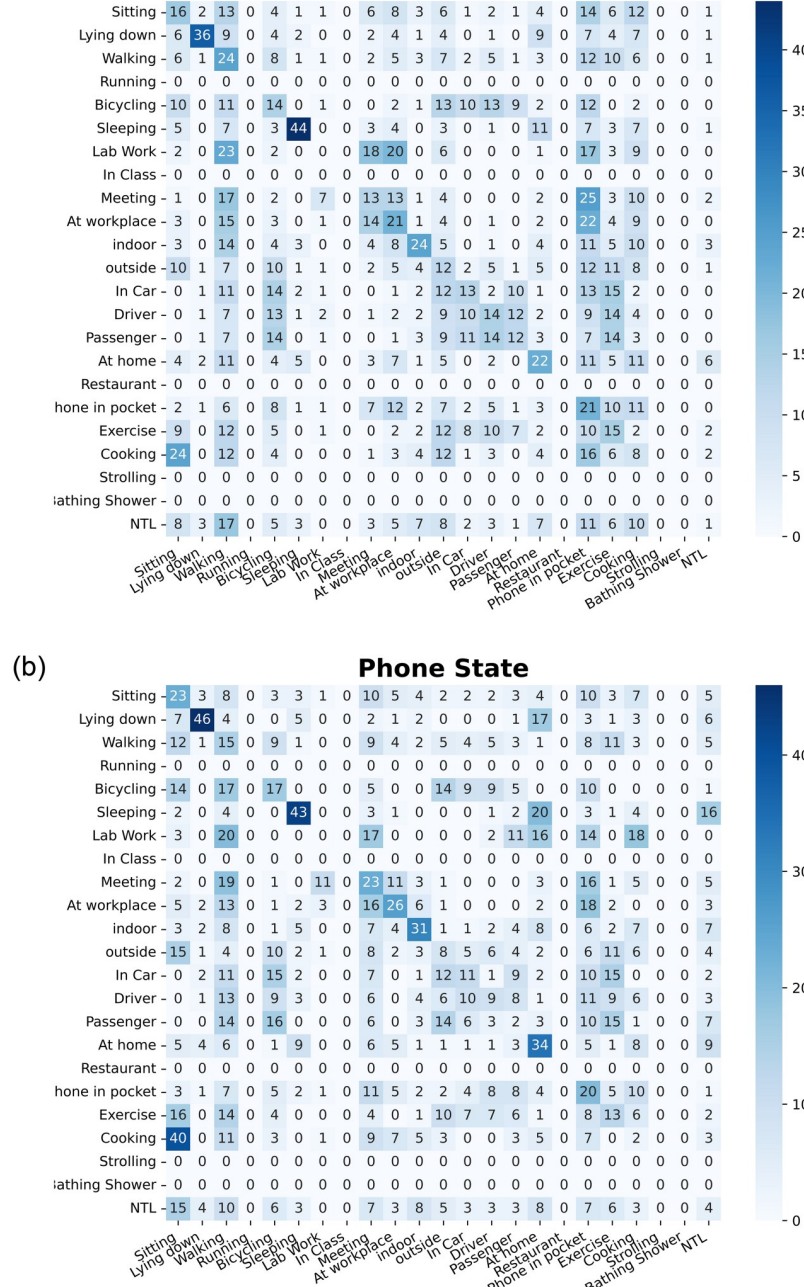

**Fig 8. Confusion matrix for location and Phone State sensors at the stabilizationtime, T = 5.**

recognizing users' behavior in a natural environment. Phone state and EF showed no or less improvement since this temporal information's existence in their data. Further, TV-6 performed better than TV-3 in all the cases. So, if we increase the temporal slots by more than six, there might be a chance of further improvement in the BA. Therefore, improved BA shows that users' behavior depends more on temporal data, i.e., part of the day, e.g., sleeping is mainly done at night or late night a day. Similarly, in the context of working people, driving is primarily done in the morning for departure or in the evening to return home. This

correlation of temporal data with users' behavior is significantly captured in improved BA. Moreover, as a comparison of different window sizes, we can clearly see that greater window size might capture more information that shows better BA than the 20*s* window.

The other attribute of the proposed methodology is the reduction of annotation load achieved using *DBQS*. *DBQS* with TV-6 reduces the annotation load for accelerometer, audio, location, and gyroscope by 6%, 18%, 8%, and 46% respectively to achieve the BP. Compared to base line's CP, *DBQS*, along with TV-6, reduces the annotation load for accelerometer, audio, phone state, and location by 6%, 8%, 18%, and 4% respectively. Both the discussed cases of annotation reduction proved that *DBQS* successfully captured the users' similar behavioral patterns in the natural environment. But watch accelerometer showed no annotation reduction due to many variations in sensor data. The reason behind this variation is due to the frequent hand motion while performing any activity. We need an extensive dataset as Extrasensory to get the advantage of annotation reduction from our *DBQS* approach based on the dissimilarity function. A more extensive dataset has higher chance of similar behavioral patterns of different users in the natural setting, and the same practice is also observed by the other study [77].

Lastly, setting stabilization wait time $T$ = 4, 5, 6 shows the convergence of the AL approach. For all the values of $T$, our proposed method converges gyroscope and watch accelerometer with more training data than the baseline method. Similarly, location and watch accelerometer showed less significant annotation load reduction. This may be due to the fact that in natural setting, users change their location and move their hands more frequently. Therefore, their sensor input data can be further processed using any clustering technique to analyze the impact of *DBQS*.

Due to the natural behavior of users' activities in real life, the considered dataset is highly skewed towards negative classes. For example, the dataset contains 34% data for *sitting* activity and 66% data for *not sitting* activity. Similarly, *cooking* activity has 1% data and 99% data for *not cooking*. This highly imbalanced data is challenging in the correct recognition of users' activities in the natural environment. State-of-the-art AL approaches only work for the informativeness of samples. However, the proposed *DBQS* method also focuses on the class distribution while querying the samples in each successive batch. In each successive batch, the method maintains the same class ratio as of whole training data and performs well while recognizing the user behavior in the natural environment.

It is demonstrated from the results of implemented methodology that with less amount of training data, we can achieve almost CP to recognize human behavior in diverse settings. Therefore, the proposed method works well than the baseline by reducing the number of samples with the help of *DBQS* and also increases BA when temporal data is included. *DBQS* with temporal information significantly affects user behavior recognition in the natural environment.

## Conclusion

In this research, we used AL to create a HAR model that can accurately predict users' activities in natural settings with low labeled data requirements to reduce annotation effort. We proposed a new query strategy, *DBQS* that works in a pool-based setting. *DBQS* maintains diversity and reduces redundant instances in each batch using the dissimilarity function between current and already selected batches. In each iteration, the query strategy samples the most informative and dissimilar samples in a batch. Along with the query strategy, we also investigated the impact of temporal information on users' behavior and activities. We showed via experimentation on the Extrasensory dataset [12] that proposed *DBQS* and temporal

information can effectively and significantly recognize user behavior. To achieve the Best Performance, models trained on accelerometer, audio, location, and gyroscope sensor using *DBQS* and temporal information, reduce annotation load by 6%, 18%, 8%, and 46% respectively. We also reported that our method converges earlier than the baseline.

In HAR, computational cost (in terms of memory and time) is another main challenge of mobile applications that needs to be focused. In our proposed methodology, we used a computationally expensive dissimilarity function, thus in the future we will work to make it less costly. We will also merge temporal and location data to analyze the impact on users' natural behavior. We will explore more variations of temporal information like working and non-working days. Further, we will work on deep learning models in future.

## Author Contributions

**Conceptualization:** Atia Akram, Asma Ahmad Farhan.

**Data curation:** Atia Akram.

**Formal analysis:** Atia Akram, Asma Ahmad Farhan, Amna Basharat.

**Methodology:** Atia Akram, Asma Ahmad Farhan.

**Project administration:** Asma Ahmad Farhan.

**Software:** Atia Akram.

**Supervision:** Asma Ahmad Farhan, Amna Basharat.

**Validation:** Atia Akram.

**Visualization:** Atia Akram, Asma Ahmad Farhan.

**Writing – original draft:** Atia Akram.

**Writing – review & editing:** Asma Ahmad Farhan, Amna Basharat.

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
