## [Decision Letter · Decision Letter 0]

17 Mar 2023

PONE-D-23-05731Less is more: efficient behavioral context recognition using DBQSPLOS ONE

Dear Dr. Akram,

Thank you for submitting your manuscript to PLOS ONE. After careful consideration, we feel that it has merit but does not fully meet PLOS ONE’s publication criteria as it currently stands. Therefore, we invite you to submit a revised version of the manuscript that addresses the points raised during the review process. The topic is interesting, methods are well described and motivated, and the conclusions are supported by the results. However, at least one reviewer recommended reconsidering the manuscript after a major revision. I invite you to revise and resubmit your manuscript after addressing the comments below.

We look forward to receiving your revised manuscript.

Kind regards,

Luigi Borzì, Ph.D.

Academic Editor

PLOS ONE

Journal Requirements:

Reviewers' comments:

Reviewer's Responses to Questions

**Comments to the Author**

1. Is the manuscript technically sound, and do the data support the conclusions?

Reviewer #1: Yes

Reviewer #2: Yes

2. Has the statistical analysis been performed appropriately and rigorously? 

Reviewer #1: Yes

Reviewer #2: Yes

3. Have the authors made all data underlying the findings in their manuscript fully available?

Reviewer #1: Yes

Reviewer #2: Yes

4. Is the manuscript presented in an intelligible fashion and written in standard English?

Reviewer #1: Yes

Reviewer #2: Yes

5. Review Comments to the Author

Reviewer #1: Dear authors, thanks for your submission. The paper tackles an interesting problem and seems to achieve quite good results, even if based on quite a simple intuition. I think nevertheless that some aspects could be improved:

- all references to Sections are absent (e.g., you write "in Section ", but there is no section number following). -the concept of active learning is referred many times in the text from the introduction but is not clearly explained until the method section. I think this description should be anticipated in the background for a more coherent and readable introduction

-please check the acronyms, then use the full name just the first time and then only the acronym. For example the acronym definition for active learning is repeated several times in the text. Consider adding an acronym list

-table 1 is never referenced in the text

-formula 8 has a . instead of a multiplication dot

-summarize numerical results in tables, reporting balanced accuracy values in text especially page 14 is not an effective description

Reviewer #2: In this paper, the authors propose a novel behavior context recognition approach i.e., Dissimilarity-Based Query Strategy(DBQS). The approach DBQS leverages Active Learning based selective sampling to find the informative and diverse samples in the sensor data to train the model. The results are corroborative and interesting. I can recommend this paper for a publication after several main concerns are addressed.

1. I recommend the authors to divide Section 1 (introduction) into the following subsections: Background, Limits of Prior ATRs, Research motivation, Main contribution.

2. How do the window length and overlap influence the final performance? The configuration details about each benchmark dataset should be introduced

3. The confusion matrices should be added.

4. In fact, there has been high imbalance in activity benchmark datasets. It will be better if the authors could analyze or discuss how the active learning help to alleviate this issue.

5. I also recommend the authors to refer to several recent activity recognition literatures in introduction part.

a. 10.1109/TETCI.2021.3136642

b.10.1109/TIE.2022.3161812

c. 10.1109/TMC.2022.3174816

d. 10.1145/3551486

e. 10.1109/TIM.2021.3102735

6. PLOS authors have the option to publish the peer review history of their article (what does this mean?). If published, this will include your full peer review and any attached files.

Reviewer #1: No

Reviewer #2: No

---

## [Author Response · Author response to Decision Letter 0]

4 May 2023

All the comments and suggestions from reviewers and editors are highly admirable. We acknowledged all do-able comments to the best of our level. For figures requirement, we passed the images to PACE and generated .tif images. But while uploading these images, create problems. Please help us in this regard. Further for figures we have also email to figures@plos.org and waiting of their response.

---

## [Decision Letter · Decision Letter 1]

12 May 2023

PONE-D-23-05731R1

Less is more: efficient behavioral context recognition using DBQS

PLOS ONE

Dear Dr. Akram,

Thank you for submitting your manuscript to PLOS ONE. After careful consideration, we feel that it has merit but does not fully meet PLOS ONE’s publication criteria as it currently stands. Therefore, we invite you to submit a revised version of the manuscript that addresses the points raised during the review process.

We look forward to receiving your revised manuscript.

Kind regards,

Luigi Borzì, Ph.D.

Academic Editor

PLOS ONE

Journal Requirements:

Additional Editor Comments:

The article has now been evaluated.

The reviewers are satisfied with the authors' answers and have no further comments to make.

I ask you to make minor changes to the manuscript to improve the overall presentation.

Please address the comments below and make minor changes to the text.

- Please define acronyms the first time they are used. Then use the acronym. Human activity recognition is defined on Line 101 and it is used in full on Lines 135,277; DBQS is defined in the abstract, then it is redefined on

Line 659 and used in full on Line 496; Active learning is defined on Line 76 and used in full on Line 76.

- In the title, I suggest using the full form of DBQS (Dissimilarity-Based Query Strategy) instead of the acronym.

- If acronyms are used in tables or images, please define them in the caption (Table 1, Table 3).

- References should be ordered (in Line 64 they are not).

- Please review the manuscript and check for typos (e.g. "visulaize" in Line 571).

- One reviewer suggested some references to their previous published articles. However, this is not fair. Please ignore insignificant references and keep only the necessary citations (e.g. reference 20 proposes a CNN, not

an LSTM in Line 31).

- The introduction includes only a few sentences. I suggest merging the Introduction and Background sections into one (Introduction).

- Improve the quality of the figures and ensure that labels and legends are visible.

Reviewers' comments:

Reviewer's Responses to Questions

**Comments to the Author**

1. If the authors have adequately addressed your comments raised in a previous round of review and you feel that this manuscript is now acceptable for publication, you may indicate that here to bypass the “Comments to the Author” section, enter your conflict of interest statement in the “Confidential to Editor” section, and submit your "Accept" recommendation.

Reviewer #1: All comments have been addressed

Reviewer #2: All comments have been addressed

2. Is the manuscript technically sound, and do the data support the conclusions?

Reviewer #1: Yes

Reviewer #2: Yes

3. Has the statistical analysis been performed appropriately and rigorously? 

Reviewer #1: Yes

Reviewer #2: Yes

4. Have the authors made all data underlying the findings in their manuscript fully available?

Reviewer #1: Yes

Reviewer #2: Yes

5. Is the manuscript presented in an intelligible fashion and written in standard English?

Reviewer #1: Yes

Reviewer #2: Yes

6. Review Comments to the Author

Reviewer #1: Dear Authors, thanks for addressing all the issues that were highlighted during the revision process.

Reviewer #2: The authors have addressed my all concerns well in the revised version, and I can recommend this paper for a publication.

7. PLOS authors have the option to publish the peer review history of their article (what does this mean?). If published, this will include your full peer review and any attached files.

Reviewer #1: No

Reviewer #2: No

---

## [Author Response · Author response to Decision Letter 1]

21 May 2023

Thank you for good suggestions and comments on our manuscript.

---

## [Editor Report · Decision Letter 2]

26 May 2023

Less is more: efficient behavioral context recognition using Dissimilarity-Based Query Strategy

PONE-D-23-05731R2

Dear Dr. Akram,

We’re pleased to inform you that your manuscript has been judged scientifically suitable for publication and will be formally accepted for publication once it meets all outstanding technical requirements.

Kind regards,

Luigi Borzì, Ph.D.

Academic Editor

PLOS ONE

---

## [Editor Report · Acceptance letter]

30 May 2023

PONE-D-23-05731R2 

Less is more: efficient behavioral context recognition using Dissimilarity-Based Query Strategy 

Dear Dr. Akram:

I'm pleased to inform you that your manuscript has been deemed suitable for publication in PLOS ONE. Congratulations! Your manuscript is now with our production department. 

Kind regards, 

on behalf of

Dr. Luigi Borzì 

Academic Editor

PLOS ONE